# Absence of Aquaporin-4 (AQP4) Prolongs the Presence of a CD11c+ Microglial Population during Postnatal Corpus Callosum Development

**DOI:** 10.3390/ijms25158332

**Published:** 2024-07-30

**Authors:** Francisco Mayo, Lourdes González-Vinceiro, Laura Hiraldo-González, Claudia Calle-Castillejo, Ismael Torres-Rubio, Manuel Mayo, Reposo Ramírez-Lorca, Miriam Echevarría

**Affiliations:** 1Instituto de Biomedicina de Sevilla, IBiS/Hospital Universitario Virgen del Rocío/CSIC/Universidad de Sevilla, 41013 Sevilla, Spain; 2Departamento de Fisiología Médica y Biofísica, Facultad de Medicina, Universidad de Sevilla, 41009 Sevilla, Spain; 3Física Teórica, Universidad de Sevilla, Apartado de Correos 1065, 41080 Sevilla, Spain

**Keywords:** aquaporin-4, CD11c+, microglia, SPP1, osteopontin, corpus callosum, cerebellum, myelinization

## Abstract

Aquaporin-4 (AQP4) expression is associated with the development of congenital hydrocephalus due to its structural role in the ependymal membrane. Gene expression analysis of periaqueductal tissue in AQP4-knockout (KO) mice at 11 days of age (P11) showed a modification in ependymal cell adhesion and ciliary protein expression that could alter cerebrospinal fluid homeostasis. A microglial subpopulation of CD11c+ cells was overexpressed in the periaqueductal tissue of mice that did not develop hydrocephalus, suggesting a possible protective effect. Here, we verified the location of this CD11c+ expression in the corpus callosum (CC) and cerebellum of AQP4-KO mice and analysed its time course. Immunofluorescence labelling of the CD11c protein in the CC and cerebellum of WT and KO animals at P3, P5, P7 and P11 confirmed an expanded presence of these cells in both tissues of the KO animal; CD11c+ cells appeared at P3 and reached a peak at P11, whereas in the WT animal, they appeared at P5, reached their peak at P7 and were undetectable by P11. The gene expression analysis in the CC samples at P11 confirmed the presence of CD11c+ microglial cells in this tissue. Among the more than 4000 overexpressed genes, *Spp1* stood out with the highest differential gene expression (≅600), with other genes, such as *Gpnmb*, *Itgax*, *Cd68* and *Atp6v0d2*, also identified as overexpressed. Therefore, CD11c+ cells appear to be necessary for normal corpus callosum development during postnatal life, and the absence of AQP4 prolonged its expression in this tissue.

## 1. Introduction

The term “microglia” first appeared in the scientific vocabulary a century ago, when Pio del Río Hortega attempted to differentiate these glial cells from oligodendrocytes [1,2]. The description of this cell population has evolved in such a way that we currently define microglia as a type of immune cell that is located only in the central nervous system (CNS), acting in various physiological and pathological situations. In the adult brain, it fulfils the classic functions of an immune cell that continuously monitors its local microenvironment, where if it encounters an injury, it activates repair and defence mechanisms, such as phagocytosis, cell death induction, neuroprotection, axonal modulation and synapse monitoring [3]. These cells are derived from myeloid progenitors of the yolk sac that begin to infiltrate the brain around mouse embryonic day 9.5 (E9.5). The closure of the blood–brain barrier, which occurs around day 13.5 (E13.5), is proposed as the mechanism that confines microglia within the brain parenchyma [2,3].

Until recently, the role of microglia in the developing brain was unknown, and a single immune role was assumed [4]. However, numerous studies have investigated the functions that this type of cell could play in normal physiology in the brain and, in addition to intervening in inflammatory reactions and in the phagocytosis of compounds and cellular debris, a prominent role is conceived in mechanisms such as synaptic pruning and monitoring, regulation of processes such as neurogenesis, revascularisation and angiogenesis, astrocytic permeability of the brain–blood barrier (BBB), in addition to a prominent role in the myelination and regulation of oligodendrocyte development [5]. The classical concept of duality for the cellular subtype [6], represented by two characteristic phenotypes designated under the names M1 (pro-inflammatory profile) and M2 (anti- inflammatory profile), has been banished by recent studies that conceive a gradient of multiple phenotypes on the spectrum of immune type. Given this panorama, some authors have suggested the possibility of considering microglia as a community of cells, as opposed to a single cell type, so that each exhibits characteristic properties that allow it to be differentially activated and, thus, play a different physiological role [7].

The development of new transcriptomic techniques such as single-cell RNA sequencing has allowed us to discern that microglia assume many distinctive states that change over time, which can be defined by unique markers [6,7]. Microglial heterogeneity has been described in different neural regions, such as grey and white matter areas, pathological conditions, and especially during early developmental stages [8,9,10,11,12]. There is greater microglial heterogeneity during postnatal development, associated with a greater diversity of functions, compared with more homeostatic adult microglia that are only altered by pathological conditions or ageing [8,9]. Among the great population diversity of microglial cells that intervene in the postnatal development of the CNS, a subtype with relevant participation in myelination and neurogenesis is of special interest [13]. Although there is discrepancy in its nomenclature due to its description in several simultaneous studies [8,9], we can call it CD11c microglia, given that it is one of the markers that differentiate it from the other populations. CD11c is an integrin protein that is part of complement receptor 4, and *Itgax* is the gene that encodes it. In wild-type (WT) mice, the appearance of these cells is dated between postnatal days 3 and 7 (P3 and P7), with P10 being the stage at which they disappear. These cells are characterised by having an amoeboid morphology with finer processes than the branched microglial cells of the adult brain, and by their location in the axonal tracts of the corpus callosum (CC) and cerebellum, areas of high myelination [3,8,9]. These cells are metabolically active and highly phagocytic, displaying a transcriptomic profile similar to those identified in other microglial subtypes associated with neurodegeneration [14,15,16,17]. Given that the appearance of this population correlates with the beginning of myelination in the CNS, the main phagocytosed cells are the newly formed oligodendrocytes that undergo massive cell death [13,18].

Furthermore, these microglia show a strong gene signature associated with agents that intervene in neurodevelopment, such as insulin-like growth factor 1 (IGF1), which has been revealed as an important trophic factor for the survival of layer V cortical neurons during postnatal development, as well as osteopontin (OPN) and macrophage colony-stimulating factor (M-CSF), which participate in neuronal homeostasis [19]. Furthermore, IGF1 is essential for primary myelination of the axonal tracts with which these microglia are associated, given that its depletion produces more unmyelinated fibres [13].

Recently, we demonstrated that aquaporin-4 (AQP4) expression is clearly associated with the early cell differentiation process that leads to myelination in areas of white matter in mice brains, such as in the CC and cerebellum [20,21]. We also identified, for the first time, a population of CD11c+ microglia in the periaqueductal zone of AQP4-knockout (KO) animals similar to that detected by other authors in the CC and cerebellum, which is believed to be important for maintaining the myelinisation and neurogenesis of these tissues in WT mice [13]. The CD11c+ cell population found beneath the ependyma of the Sylvian aqueduct displays longer and higher expression levels in the periaqueductal zone of the AQP4-KO animal than in WT mice [22]. We propose that CD11c+ overexpression during postnatal development (P3-P11) in KO mice is necessary to reduce ependyma abnormalities that could otherwise contribute to alterations in cerebrospinal fluid circulation and potentially participate in the development of hydrocephalus. 

In the present study, we analysed whether the lack of AQP4 expression in the AQP4-KO animal would affect the time course of the appearance and contraction of this CD11c+ population in the CC and cerebellum, when the postnatal development of these tissues occurs, and compare it with the time course expression of CD11c+ cells in the same tissues of the WT animal. Importantly, we show here, as previously demonstrated for the aqueduct ependyma, that the higher expression of *Ssp1* by the CD11c+ microglia appears to be important to attain the normal development of the CC and cerebellum in AQP4-KO mice. Our findings further support the key role of AQP4 presence in the neurodevelopment of these brain tissues. 

## 2. Results

### 2.1. Comparative Time Course Analysis of the Presence of Postnatal CD11c+ Microglia in the Corpus Callosum of WT and AQP4-KO Animals

To define the temporal dynamics of CD11c+ microglial cells in the CC, we performed immunofluorescence assays against IBA1 and CD11c at various points during postnatal development (P3, P7, P11 and P20). The results obtained (Figure 1A–D) showed great similarity to what had already been reported in a previous study for this microglial subtype, detected in the peri aqueduct zone of postnatal mice at the same age [22]. An earlier presence (visible from P3) of these microglia was observed in the CC of the transgenic animal, although by P7, the density of these microglial cells increased equally in both the WT and KO animals. In the CC, as detected in the Sylvian periaqueductal area and shown in the cerebellum (Appendix A), the CD11c+ cells remained highly expressed only in the AQP4-KO animals by P11, by which time they were totally undetectable by immunofluorescence in the WT animal (Figure 1C). The quantification performed (Figure 1E) confirmed an increment in the density signal for CD11c+ cells at P7 with respect to P3, showing no significant differences between the density of the microglial signal in the CC of the WT or AQP4-KO animals at P7. At age P11, however, CD11c+ cells were undetectable in the WT animal, whereas in the AQP4-KO animal, the signal remained the same as that detected at P7 (*** *p* < 0.001). As shown in Figure 1D, these CD11c+ cells were absent from the CC at P20 in both types of animals, following a time course expression similar to that shown in Sylvian periaqueduct tissue [8] and in the cerebellum (Appendix A).

### 2.2. Comparative Transcriptomic Analysis of Corpus Callosum Tissue in WT Versus AQP4-KO Animals

To study the relevance of AQP4 in the CC during the postnatal phase, we compared WT with AQP4-KO animals, focusing on the P11 postnatal stage. At this age, as had been described in the WT animal [20], a high expression of AQP4 occurs in the CC coinciding with the intense myelination process that takes place in this tissue during the second week after birth. Just as with the periaqueductal tissue, we extracted RNA from the CCs of the WT and AQP4-KO animals (n = 3 per condition) and obtained full transcriptomic profiles with a microarray. A comparison of gene profiles revealed strong homogeneity between samples from the same animal type, as shown by the component analysis (Figure 2A). Furthermore, the differential gene expression analysis indicated significant changes according to the predefined statistical thresholds (false-discovery rate/q. value < 0.05 and with a fold change >1.5) in a total of 4953 genes (840 inhibited and 4113 overexpressed in the AQP4-KO/WT comparison, Figure 2B). Among the more than 4000 overexpressed genes, Spp1 stood out as having the highest differential gene expression (≅600). Furthermore, as in the aqueduct transcriptomic analysis [8], other genes, such as Gpnmb, Itgax, Cd68 and Atp6v0d2, were also identified as overexpressed. The study of gene set enrichment (Figure 2C) highlights the enrichment of microglial and myeloid cell profiles (normalised enrighment score > 3 in both cases), as well as a clear increase in Gene Ontology enrichment values for immune functions such as phagocytosis or adhesion and proliferation of leucocytes, among functions associated with a strengthening of immune processes. 

### 2.3. Differential Expression of Genes Associated with Immune Processes and Identification of Cellular Origin of the Ssp1 Transcript 

To confirm the changes indicated by the transcriptomic study of microglial genes, validations were performed by qPCR on biological samples independent of those used in the microarray study. The results obtained (Figure 3A) highlight the secreted phosphoprotein 1 gene (Spp1), which codes for the osteopontin protein, as the gene with the greatest overexpression, with expression levels approximately 535-times those obtained in the WT tissue (*p* < 0.01). Genes such as Gpnmb and Itgax also showed increments in their expression levels within a similar range, and other genes, including Igf1, Cd68, Lgals3, ApoE and Trem2, experienced significant increases, although of lesser magnitude.

To determine the cellular origin of the Spp1 transcript detected in the CC of the AQP4-KO animals, we performed in situ hybridisation assays in mice at P11. The results indicated an exclusive presence of Spp1 mRNA in the tissue of the AQP4-KO animal (Figure 3B). In the higher-magnification image (Figure 3B’), its location was confirmed inside cells with rounded amoeboid morphology, poor in branches, very similar to those observed in the cerebral aqueduct for animals at the same stage [22]. 

Next, we performed immunofluorescence assays against SPP1 and CD11c (using anti-osteopontin and anti-CD11c primary antibodies, respectively) to corroborate the colocalisation of both upregulated markers in the same cell type. As reflected in the confocal microscopy images (Figure 4A,B), the CD11c+/SPP1+/Iba1+ microglial presence was identified only in the CC of the AQP4-KO animal at P11. Micrographs taken at higher magnification (Figure 4C) highlight the colocalisation of markers on cells located in the CC (yellow arrows), as opposed to those located in regions closer to cortical areas that showed unique specificity against Iba1 labelling (white arrows). Employing three-dimensional modelling (Figure 4D), we confirmed its amoeboid structure, reinforcing the similarity with the microglial type identified in the Sylvian aqueduct [22]. 

### 2.4. Comparative Analysis of the CD11c+ Microglia in the Corpus Callosum with Other Microglia

Lastly, we evaluated the degree of similarity of the gene profile of the microglia identified in the CC with that of the microglia found in the Sylvian aqueduct of P11 mice and other reported microglial populations [8,9,22]. The results of the analysis (Figure 5A) demonstrated a strong homology between the microglial profile identified in the CC (AQP4-CC) and that of the microglial subtypes PAM and CD11c; however, the homology was even higher with the microglia signature identified in the Sylvian aqueduct zone (AQP4-AC) of the same AQP4-KO animals (correlations of 0.64, 0.71 and 0.89, respectively). Consequently, the analysis highlighted a greater degree of similarity with the microglial subtypes linked to neurodevelopment (CD11c and PAM) than with the subtypes that emerged in response to a pathological situation (DAM, APP, MGnD or LDAM subtypes) or ageing (Aged).

This proposal was further supported by transmission electron microscopy (TEM) observations of the CD11c+ cellular ultrastructure (Figure 5B), which highlighted the high presence of inclusion bodies possibly associated with intense phagocytic activity. Images at high magnification revealed enrichment in the cellular inclusion bodies (phagocytic vesicles), a fundamental characteristic that has been described for CD11c, PAM and AQP4-Aq microglia, all linked to neurodevelopment [13,14].

## 3. Discussion

Our most recent studies have demonstrated that AQP4 expression is clearly associated with myelination in areas of white matter, as in the CC and cerebellum in the brains of mice [20,21]. Additionally, we identified a population of CD11c+ microglia in the mouse periaqueductal zone [22], similar to the one detected by other authors in the CC and cerebellum, which had been reported to be very important in maintaining these tissues’ myelinisation and neurogenesis in WT mice [13]. A recent study characterized the time course of CD11c presence in different neural regions [23], shedding light on the presence of this essential microglial cell subtype during the neurodevelopmental context. Interestingly, we observed that this CD11c+ microglial cell population remained in the periaqueductal area of the AQP4-KO animals for longer and at higher levels than those detected in the WT animals, allowing us to propose that their overexpression during postnatal development in KO mice was necessary to reduce abnormalities of the ependyma and prevent the possible development of hydrocephalus.

If we examine the seminal studies performed by del Río-Hortega, they described the existence of amoeboid microglial cells, preferentially located in regions of white matter during neonatal development, a cell population for which the author coined the term “source of microglia” [5]. Recent studies have characterised these microglia, highlighting their course of appearance between points P5 and P7, a time frame consistent with that found in our study in WT animals (Figure 1). In AQP4-KO animals, however, their presence manifested earlier, by P3, and expanded until P11 (Figure 1), at considerably high levels of expression.

The former demonstration of the exacerbated expression of AQP4 in the CC during the perinatal period [20] motivated us to delve into the consequences that congenital deletion of AQP4 could generate in the development of this tissue. Here, using diverse experimental approximations, such as immunofluorescence, microarray analysis (Affymetrix, Santa Clara, CA, USA), RT-qPCR and in situ hybridisation, we came to the same result, which is the confirmation that a CD11c+ microglial cell population appears and remains on the CC tissue of the AQP4-KO animals for a longer time and at higher levels than those detected in the CC of WT animals. Remarkably, the results of the comparative transcriptomic study highlighted the gene *Spp1*, which codes for osteopontin, as well as the rest of the genes associated with the microglial profile of the identified subependymal subpopulation, such as those that showed greater differential overexpression in the CC of the AQP4-KO animals (Spp1, Gpnmb, Igf1, Itgax, among others) (Figure 2). The significance of these increases was robustly validated by qPCR in independent samples (Figure 3), and, via in situ hybridisation (Figure 3B,B’) and an immunohistochemical approach (Figure 4), the presence of CD11c+ cells with a typical amoeboid morphology was confirmed as that responsible for the overexpression of Spp1. 

The analysis of similarities between the transcriptomic signatures of this microglia and those reported by previous studies led us to establish a correlation matrix (Figure 5A) that shows a solid genetic similarity with the microglial subtype identified in our mice (AQP4-CC), in the present work, and the microglial types characterised in the postnatal neurodevelopment of white matter regions such as in CD11c and PAM (with correlation levels of 0.71 and 0.64, respectively). However, the maximum correlation value reached was obtained with the subependymal microglia of the aqueduct (AQP4-AC) reported previously [22], a result that could indicate the presence of the same microglial type in both tissues (aqueduct and CC) of the AQP4-KO animals used in our laboratory (AQP4-KO/C57BL/6). The great similarity in the transcriptomic signatures of both microglia could also be attributed to an overestimation due to the shared genetic background of the animals used or to the use of the same transcriptomic analysis method (microarray), as opposed to RNA-seq studies developed by the rest of the groups included in the transcriptomic signature comparison shown in Figure 5. The degree of homology exhibited with populations related to pathological situations (DAM1 and DAM2, APP and LDAM subtypes) or ageing (Aged) was lower. To provide additional evidence of the association between these microglia and neurodevelopment (rather than with disease), we analysed the cellular ultrastructure of these microglia. Using TEM, we detected the accumulation of gold particles in cell subtypes that, due to their location (CC area) and amoeboid morphology, could correspond to the identified microglial type. The ultrastructure images obtained in the present study showed high levels of phagocytic inclusion bodies (phagocytic vesicles), a fundamental characteristic that has been reported for CD11c and PAM microglia, with both microglia linked to neurodevelopment [13,14]. 

In the CC, the presence of this CD11c+ microglial type, with high phagocytic capacity during a narrow window of postnatal development, would help the tissue maturation process through phagocytosis of both the oligodendrocyte precursor cells [24,25,26] and recently formed oligodendrocytes [8] produced in excess during the myelination process. Several studies have explored the connection between both processes [26,27,28,29], suggesting several mechanisms of activation for these cells, as well as their essential role after demyelinating damage. In addition to this function, its ability to release factors with pro-myelination potential, such as osteopontin [30], galectin-3 [31] and especially igf1 [4,26], would make the CD11c+ microglial population fundamental to achieving normal levels of axon myelination in the CC and cerebellum and, thus, encourage normal development of these tissues, in agreement with similar results reported before [26]. We hypothesised that alterations leading to affecting the access of this microglia to the correct place at the precise moment during tissue development would intervene in processes of special relevance to myelination, such as promotion, proliferation and correct location of oligodendrocyte precursor cells in tissue, thus affecting the maturation of oligodendrocytes and myelination of the tissue integrated into the white matter such as the CC and cerebellum. In line with this, experiments using drugs as BLZ945 [26] that disrupt or deplete the access of CD11c+ microglia to these white matter tissues would help to further explore the functional role of this neonatal CD11c+ microglia type in the myelination process in our AQP4-KO animals. Furthermore, the specific analysis correlating the presence of these neonatal microglia with the presence of precursors (OPC) and mature oligodendrocytes in CC and cerebellum of the AQP4-KO animals would be of high interest to continue deciphering the molecular bases that bind the AQP4 expression with CNS disorders such as Alzheimer’s, hydrocephalus or NMO, among others. 

In summary, we identified a neonatal CD11c+ population that appears necessary for normal development of the CC by providing factors such as osteopontin during a short window of the neonatal period. The congenital lack of AQP4 expression prolonged the expression of this CD11c+ microglial population in this tissue, as well as in the cerebellum (Appendix A), assuring, in the end, an “apparently” normal development of these white matter areas of the CNS.

## 4. Materials and Methods

Animal care: Wild-type C57BL/6 and AQP4-KO mice were housed in a regulated temperature environment (22 ± 1 °C) with a 12-h light/dark cycle and ad libitum access to food and water. AQP4-KO mice were genotyped as previously described [32]. For euthanasia, the mice received terminal anaesthesia induced by a combination of 100 mg/kg ketamine (Pfizer, New York, NY, USA) and 10 mg/kg xylazine (Bayer, Leverkusen, Germany). All experiments were performed according to the European Directive 2010/63/EU and the Spanish RD/53/2013 on the protection of animals used for scientific purposes. Animal procedures were approved by the Animal Research Committee of the Virgen del Rocío University Hospital (26/01/2017/017; University of Seville). 

### 4.1. Immunohistochemistry and Imaging Quantification

To conduct histological analyses, mice at postnatal days 3, 7, 11 and 20 (P3, P7, P11 and P20) were anaesthetised and intracardially perfused with phosphate-buffered saline (PBS, Sigma, St. Louis, MO, USA) followed by 4% paraformaldehyde (PFA, Sigma) in PBS. Subsequently, the brains were promptly removed, fixed in 4% PFA in PBS for 2 h at 4 °C, then cryoprotected in 30% sucrose in PBS at 4 °C for 24 h and embedded in optimal cutting temperature compound (O.C.T. Compound, Tissue Tek, Electron Microscopy Sciences, Hatfield, PA, USA) before storage at −80 °C. Coronal sections of 30 µm thickness were obtained using a cryostat (Leica, Wetzlar, Germany) and mounted on Superfrost Plus slides (Thermo Fisher Scientific, Waltham, MA, USA) for their use. The sections were washed twice with PBS, permeabilised with 0.1 M PBS with 0.3% (*v*/*v*) Triton (Sigma) (PBT-0.3%) and washed with PBT-0.1% twice before adding the blocking solution (10% goat or horse serum [Sigma] and 1 mg/mL bovine serum albumin [Sigma] in PBT-0.1%), for 1 h in incubation chambers (Electron Microscopy Sciences). Sections were then incubated with the following primary antibodies in the blocking solution overnight at 4 °C: anti-osteopontin (1:100, Santa Cruz, CA, USA), anti-CD11c (1:500, Bio-Rad, Hercules, CA USA) and anti-Iba1 (1:500, Wako, Monza, Italy). The next day, the sections were washed 4 times with PBT-0.1% and incubated with the secondary antibody in the blocking solution for 2 h in dark conditions (anti-rabbit IgG conjugated with Cyanine CyTM3 [1:200, Jackson Immunoresearch, West Grove, PA, USA); anti-Hamster Armenian IgG conjugated with Alexa Fluor488 [1:400, Abcam, Cambridge, UK); and anti-mouse conjugated with Alexa Flour635 [1:400, Invitrogen, Waltham, MA, USA). The nuclei were stained with 4′,6-diamidino-2-phenylindole (1:1000, Sigma), and tissue sections were mounted with Dako fluorescence mounting medium (Dako, Glostrup, Denmark). 

Colocalisation images were acquired with a Leica Stellaris 8 confocal microscope. In all cases, defined areas were established in the histological sections, and several layers were acquired by reading on the Z axis with a 0.7 um Z-stack series. To quantify the abundance of the CD11c protein in the coronal sections, the optical density of the CD11c-positive staining was measured with NIH Image software (ImageJ, version 1.53t, NIH Image, Bethesda, MD, USA). For this study, 3 slices per mouse were included, representative of the cerebral region comprised between +0.62 and −0.10 mm in Bregma coordinates for the CC region, and between −5.68 and −6.64 mm for the cerebellum. After defining the properties for a specific region of interest for each region, the percentage of CD11c-positive fluorescence area was measured.

### 4.2. In Situ Hybridisation (RNAscope)

The brains of the mice at P11 were fixed and processed as previously described, cryoprotected in 30% sucrose in PBS, and embedded in O.C.T. (Tissue Tek, Electron Microscopy Sciences, Hatfield, PA, USA). Coronal brain sections of 20 µm thickness were obtained with a cryostat (Leica, Wetzlar, Germany) and mounted on Superfrost Plus slides (Thermo Fisher Scientific). Bregma sections from +0.62 to 0.10 mm were selected for visualisation of the CC. In situ hybridisation using the RNAscope^®^ (Bio-Techne, Minneapolis, MN, USA) technique was conducted following the manufacturer’s instructions (advanced cell diagnostics [ACD]) for fixed frozen tissue sections. Antigen retrieval and protease treatment were performed according to kit instructions (RNAscope H_2_O_2_ & Protease Plus ACD #322330). The Spp1 probe (ACD #435191) was hybridised for 2 h followed by 6 amplification steps with a HybEZ oven (ACD). Signal detection was achieved with the Brown RNAscope 2.5 HD detection kit (ACD #322310), with an incubation time of 10 min. Slices were mounted with Fluoromount-G mounting medium (Invitrogen, Waltham, MA, USA), and images were obtained using a wide-field inverted microscope with a Leica DMi8 and THUNDER computational clearing software.

### 4.3. Transmission Electron Microscopy and Immunogold

For this method, the mice underwent transcardial perfusion with a 2.5% glutaraldehyde solution in cacodylate buffer (0.1 M, pH 7.5). Later, the brains were extracted and postfixed for 2 h in the same solution. After this, tissue sectioning was performed by vibratome (Leica VT1000M, Leyca Byosistem, Wetzlar, Germany). For the immunogold assays, sections of interest underwent PBS washing followed by immersion in a 50 mM glycine solution for over 10 min. Following the standard protocol previously described [22], the samples were then exposed to polyclonal anti-IBA1 (1:50; Wako) antibodies at 4 °C overnight. After PBS rinsing, the sections were treated with a 1.4 nm gold AF488 conjugated secondary antibody (1:25, Nanoprobes, Yaphank, NY, USA) for 2 h at room temperature. The samples were later fixed again in a PBS buffer containing 1% glutaraldehyde at room temperature and then subjected to enhancer reagents (Goldenhance™ EM Formulation, Nanoprobes) for 5 min to amplify the signal. Lastly, the samples were rinsed, postfixed in 2% osmium tetroxide, and embedded in epoxy resin. Ultrathin sections were cut and examined with an electron microscope (Zeiss Libra 120, Zeiss, Oberkochen, Germany).

### 4.4. Microarray-Based Transcriptomic Analysis

For transcriptomic examination of the CC, the ClariomSTMMicro Assay mouse kit (Thermo Scientific; Affymetrix) was utilised. Each condition comprised 3 samples, each representing a pool of 3 biological replicates. RNA integrity was assessed using capillary electrophoresis with the Agilent 2100 Bioanalyzer. Subsequently, RNA preamplification was conducted with the GeneChip^TM^ IVT Pico kit (Thermo Fisher Scientific) following the manufacturer’s standardised protocol. Of the resulting cDNA, 5 μg underwent fragmentation and labelling for hybridisation in a Clariom^TM^ mouse assay (Thermo Fisher Scientific). This was followed by washing, labelling (GeneChip^TM^ Fluidics Station 450; Thermo Fisher Scientific) and scanning (GeneChipTM Scanner 3000; Thermo Fisher Scientific). Data pre-processing, including background correction, normalisation and summarisation, was executed using the Robust Multichip Average method. We employed the BrainArray annotation library pd.clariomsmouse.mm.entrezg (version 25.0.0) for mapping probes to genes. Differential expression analysis was performed with the limma package (version 3.46.0) [30]. A gene cluster enrichment analysis was conducted to extract biological insights from the list of differentially expressed genes [31], utilising gene sets from the Signature Database (v7.4). Lastly, volcano-type analysis graphs and a heatmap were generated using Transcriptome Analysis Console software (Affymetrix, Thermo Fisher Scientific).

### 4.5. RNA Extraction and Quantitative Reverse-Transcription Polymerase Chain Reaction Analysis

The CC regions from the P11 animals were microdissected in ice-cold diethyl pyrocarbonate-PBS under a stereoscopic binocular microscope (Olympus SZX16, Tokyo, Japan). Total RNA was isolated using the RNeasy Micro Kit (Qiagen, Hilden, Germany) according to the manufacturer’s instructions. The cDNA synthesis was performed using the QuantiTect reverse-transcription kit (Qiagen, Hilden, Germany), and relative mRNA expression levels were measured using quantitative real-time polymerase chain reaction (RT-qPCR) with a ViiA 7 real-time PCR system (Thermo Fisher, Waltham, MA, USA). RNA expression levels were normalised (using 18S ribosomal mRNA), and all samples were analysed in triplicate. Primer Express software v2.0 (Applied Biosystems, Waltham, MA, USA) was used to design all primers used. RNA and cDNA quality and quantity were assessed with a NanoDrop ND-1000 UV–vis spectrophotometer (Thermo Fisher).

### 4.6. Comparative Transcriptomic Analysis of the Microglial Population

Following a methodological procedure previously described [22], datasets generated from various studies [8,13,14,15,16,33,34,35,36] were accessed and used for the comparison. The microglial subtypes we analysed were proliferative region-associated microglia (PAM); axon tract-associated microglia (ATM); CD11c; microglia isolated from the brain of an Alzheimer’s animal model, the amyloid precursor protein (APP); microglia isolated from the brain of aged mice (Aged); type 1 and 2 disease-associated microglia (DAM1 and DAM2); neurodegenerative microglia (MGnD), lipid-droplet-accumulating microglia (LDAM) and a novel microglial subtype that we identified in the cerebral aqueduct of the AQP4-KO mice at the postnatal stage (AQP4-AC) [22].

To compare the different population, a clustering algorithm that uses the Euclidean metric is used [37,38,39,40,41]. This kind of analysis is known as Multidimensional Scaling (MDS), and it is used to translate distances between each pair of objects in a set into a configuration of points mapped into an abstract Cartesian space [42]. Note that MSD is a powerful tool to analyse the gene expression and is widely used with this purpose [43,44,45,46]. In order to clarify the concrete algorithm that was used in this work, a concise explanation of the method is developed below. Considering the population set as n list of expression profiles with $n^{\prime }$ attributes corresponding with gene expressions, a similarity matrix was defined: (1)$$D=\left(\begin{array}{ccc} d_{1,1} & \cdots & d_{1,n}\\ \vdots & \ddots & \vdots \\ d_{n,1} & \cdots & d_{n,n} \end{array}\right),$$
in which $d_{i,j}$ represents the Euclidean distance between the vector population $j=(i_{1},\ldots ,i_{n^{\prime }})$ and $j=(j_{1},\ldots ,j_{n^{\prime }})$, which is defined as
(2)$$d_{i,j}\;\equiv \;\sqrt{\sum_{k=1}^{n^{\prime }}{\left(i_{k}-j_{k}\right)}^{2}.}$$

The matrix D, Equation (1), gives similarity based on a positive linear correlation between expression profiles, which can detect similar or identical regulation [8]. Thus, the closer the distance between populations, the greater their similarity. Moreover, we determined the cross-correlation matrix as follows:(3)$$C=\left(\begin{array}{ccc} c_{1,1} & \cdots & c_{1,n}\\ \vdots & \ddots & \vdots \\ c_{n,1} & \cdots & c_{n,n} \end{array}\right),$$
in which $c_{i,j}$, is the projection of vector i over j
$$c_{i,j}\equiv \;|\cos\theta |\;=\frac{|i\cdot j|}{\;||i||||j||}$$
and $i\cdot j\;\equiv \;\sum_{k}i_{k}j_{k}$ is the Euclidean scalar product, $||i||\;\equiv \;\sqrt{\sum_{k}{i_{k}}^{2}}$ is the magnitude, and *θ* is the angle between the vectors. It is important to note that $c_{i,j}\in \left[0,1\right]$. Thus, the elemental values of C, Equation (3), represent a percentage value of how the compared populations are correlated. A zero value indicates a total absence of relationship between the compared populations. Vice versa, the unity implies that the populations are completely related so are equivalent. Matrices (1) and (3) are represented with python software, indicating a relationship between the populations.

### 4.7. Statistical Analysis

The data are presented as mean ± standard error of the mean (SEM), and the statistical test performed is indicated in each figure legend. For all the analyses performed, normality was checked using the D’Agostino and Pearson test or the Shapiro–Wilk test. When this was confirmed, a variance analysis was performed with Tukey’s HSD post hoc analysis for multiple group comparisons or Student’s *t*-test (for 2-group comparisons); otherwise, the nonparametric Kruskal–Wallis H test (for multiple comparisons) or the Mann–Whitney U test (for 2-group comparisons) was used. GraphPad Prism Software (8.4.2 version, San Diego, CA, USA) was employed for the statistical analysis and graph design.

## Figures and Tables

**Figure 1 ijms-25-08332-f001:**
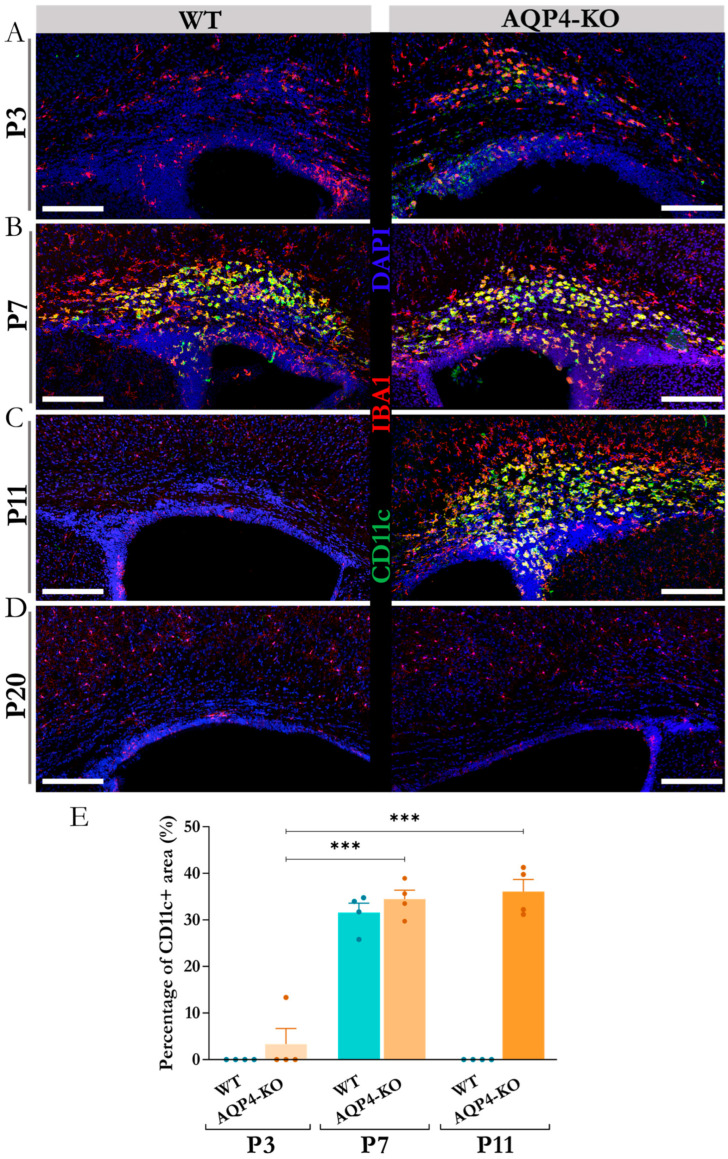
Temporal course of appearance of the CD11c+ microglia in the corpus callosum during early postnatal development. Micrographs were obtained via confocal microscopy to analyse the presence of CD11c+ microglial cells comparatively (WT vs. AQP4-KO) at postnatal stages P3 (**A**), P7 (**B**), P11 (**C**), and P20 (**D**). Scale bars = 150 μm. (**E**) Quantification of the CD11c+ microglial abundance in the analysed histological sections. N = 4 samples per group. Significant differences between groups were assessed using a one-way ANOVA followed by Tukey’s post hoc test (*** *p* < 0.001).

**Figure 2 ijms-25-08332-f002:**
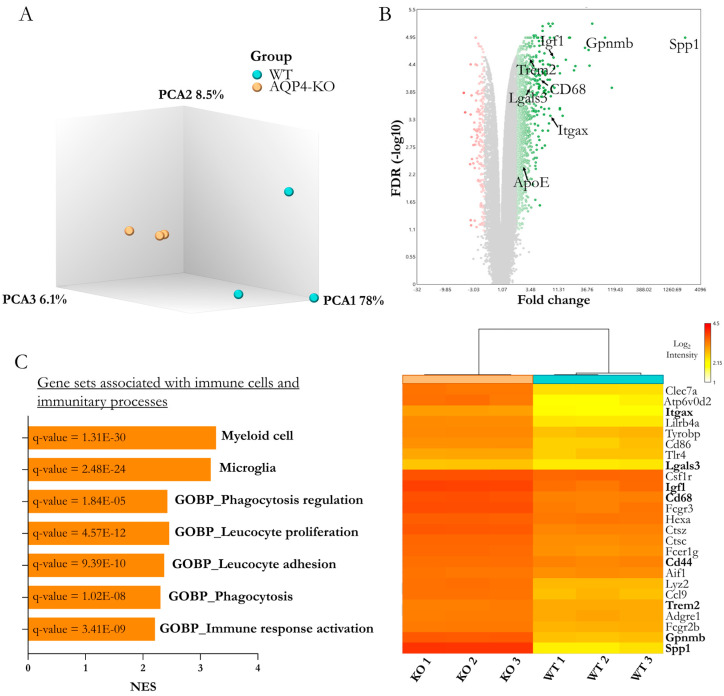
Comparative transcriptomic microarray-based analysis of the corpus callosum of WT and AQP4-KO mice at developmental stage P11. (**A**) Principal component analysis (PCA) of the transcriptomic profile of the evaluated samples (N = 3 per group). (**B**) Volcano plot includes the 4953 genes significantly expressed by FDR/q value < 0.05 (in green gradient according to the degree of overexpression, or red gradient for those identified as inhibited). (**C**) Gene sets associated with immune function showed an overexpression in the AQP4-KO corpus callosum. On the right side, the expression levels of different genes included in these gene sets are represented in the heatmap. Bold genes in the heatmap and highlighted genes in the Volcano plot were genes validated by qPCR (Results shown in Figure 3).

**Figure 3 ijms-25-08332-f003:**
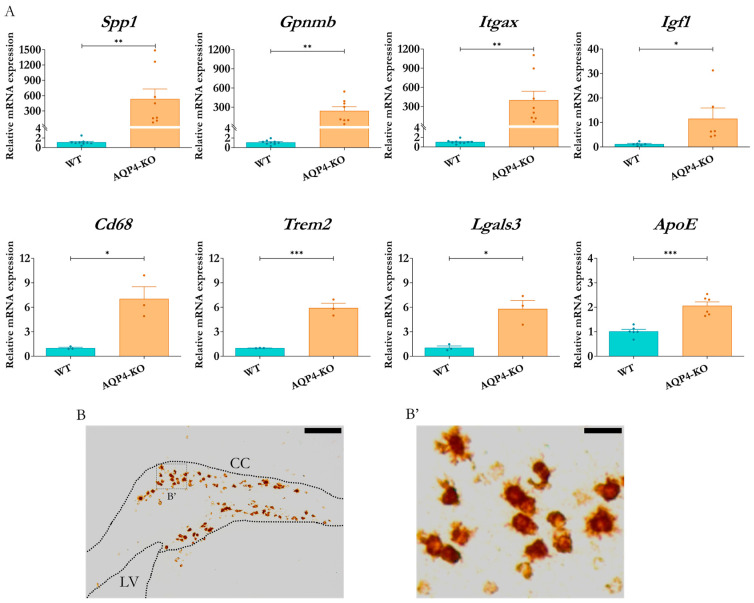
Analysis of differential expression of genes linked to immune populations and processes. (**A**) RT-qPCR validations of expression levels in the corpus callosum (AQP4-KO and WT to P11) for genes associated with the previously characterised postnatal microglial subtype that were identified as significantly expressed by microarray. N = 3–9 per condition. Unpaired Student’s *t*-test for significance analysis. * *p* < 0.05; ** *p* < 0.01; *** *p* < 0.001. Identification of Spp1 mRNA by ISH in coronal brain sections (LV = lateral ventricle, CC = corpus callosum) from AQP4-KO animal at P11 (**B**). (**B’**) Higher-magnification image of the structure showing the presence of Spp1 mRNA in amoeboid-like cells (**B’**). Scale bars = 150 µm (**B**) or 20 µm (**B’**).

**Figure 4 ijms-25-08332-f004:**
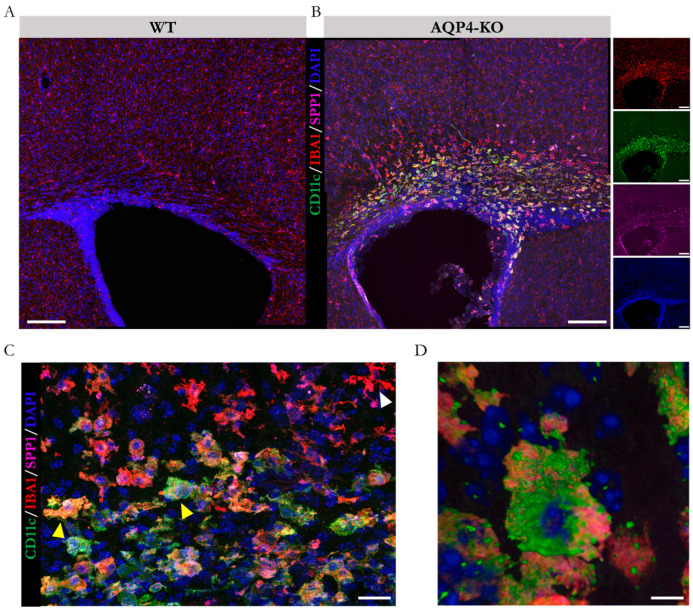
Determination of the presence of the microglial subtype CD11c as the source of Spp1 overexpression. Micrographs of immunofluorescence staining against the markers SPP1 and CD11c (overexpressed at the gene level), as well as against the pan-microglial marker IBA1 in coronal brain sections from WT (**A**) and AQP4-KO (**B**) animals at P11. For better visualisation of the markers in (**B**), they are depicted separately on the right. (**C**) Higher-magnification micrograph of the corpus callosum region with cells positive for all 3 markers (yellow arrows) or IBA1 alone (white arrow). (**D**) Three-dimensional reconstruction of a representative example of the identified CD11c+ amoeboid microglial CD11c+ cell. Scale bars = 150 µm (**A**,**B**), 20 µm (**C**), and 5 µm (**D**). N = 7–8 animals per condition were analysed (4 sections per animal).

**Figure 5 ijms-25-08332-f005:**
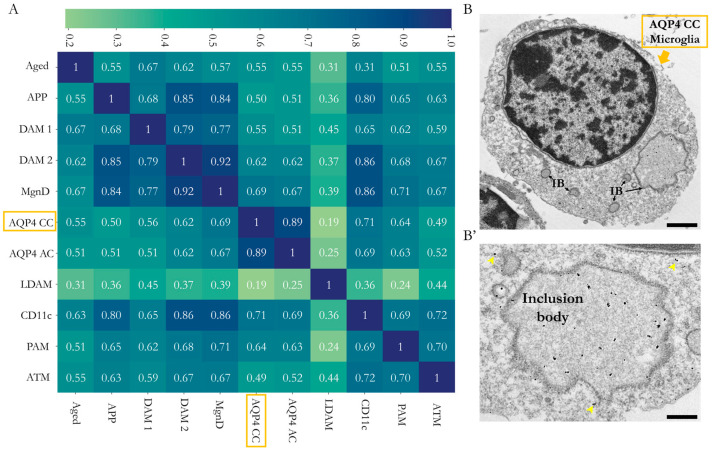
Comparative study of the microglial gene profile identified in the CC of the AQP4-KO animals and characterisation of its ultrastructure. (**A**) Correlation matrix of the gene profile constructed from the data obtained by microarray and those provided by the studies that have defined various microglial types by transcriptomic approaches. (**B**) Electron micrographs of sections of corpus callosum subjected to immunolabelling assay by immunogold against IBA1. In the magnified image (**B’**), the presence of gold colloidal particles (yellow arrows), as well as cellular structures (IB or inclusion body), is marked. Scale bars = 1 µm (**B**) or 350 nm (**B’**).

## Data Availability

The data presented in this study are available in: https://idus.us.es/browse?type=author&value=Echevarr%C3%ADa+Irusta%2C+Miriam (accessed on 18 February 2023).

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
