# Peer review of "Absence of Aquaporin-4 (AQP4) Prolongs the Presence of a CD11c+ Microglial Population during Postnatal Corpus Callosum Development"

_ijms, 2024, doi:10.3390/ijms25158332_

Round 1

Reviewer 1 Report

Comments and Suggestions for Authors     The manuscript presented from Francisco Mayo et al., entitled  "Absence of AQP4 prolongs the presence of a CD11c+ microglial population during postnatal corpus callosum development" is interesting and original. However the authors must improve the introduction also including more reference (only 6 reference in this version of the manuscript, very low). Microglia is a hot topic in the scientific community. The authors should check the statistical analysis of two figures, 4 and 5, I have different concerns. The quality of the images is good but the size is really low, they should improve the magnification. The discussion must be improved. They showed different results using more experimental approaches, however the discussion is poor.      

Author Response

Reviewer 1.

We wish to thank this reviewer for the time she/or he spent to evaluate our paper and for her/his comments that serve us to clarify some aspects of our work and make more complete. We have taken into consideration her/his suggestions and make some modifications that we think improve the quality of the manuscript. We hope those changes will satisfy most of the points raised by the referee, and therefore make the paper adequate for publication.

According to this Reviewer: “the authors must improve the introduction also including more reference (only 6 reference in this version of the manuscript, very low). Microglia is a hot topic in the scientific community. The authors should check the statistical analysis of two figures, 4 and 5, I have different concerns. The quality of the images is good but the size is really low, they should improve the magnification. The discussion must be improved. They showed different results using more experimental approaches, however the discussion is por”.

Answer: Following the suggestions indicated by this reviewer we introduced several changes in the manuscript as resumed below:

1.- In the Introduction, three new paragraphs were included to improve this section of the manuscript:

- the first goes between lines 46 and 60

- next, between lines 63 and 65

- and the third, between lines 80 1nd 81.

  1. In the legend of figure 4, the number of animals and sections used per animals were added. This figure 4 does not contain numerical data, qualitative observations are enough to see the difference between both conditions, but the statistical analysis for the comparison of CD11c+ cells between WT and AQP4-KO animals would be equivalent or correspond to the analysis shown in Figure 1E, at P11.

  1. In the Discussion, several paragraphs were added to enrich this section of the manuscript:

- the first goes between lines 275 and 277

- next, between lines 293 and 298

- next, a sentence in lines 301-302, and 303

- next, a sentence in line 310

- next, a sentence in line 315

- next, a paragraph of 3 lines between lines 334-336

- and finally, a paragraph of 8 lines between lines 346-353.

  1. In the Materials and Methods section, also a new paragraph was inserted to give a more complete description of the statistical method used to generate the Figure 5.

- the paragraph goes between lines 473-480

  1. All figures will be load individually in the platform of the journal, instead of imbedding them in the text, because transforming the file to pdf make the figures of low quality.

New references (19) were added to the manuscript along the different sections of it. We agree with the Reviewer that microglia are a very hot topic and many of these new references are solid recent papers that make stronger and give extra support and relevance to our findings. We truly appreciate the reviewer indication for encouraging us in this sense to improve our work.

Reviewer 2 Report

Comments and Suggestions for Authors

Francisco Mayo et al. reported on the distribution of microgia, which are CD11c-positive cells, depending on the presence or absence of AQP4. AQP4 is closely related to diseases, including its function in blood vessels and various brain regions, and is therefore considered to be a very significant study.

There seems to be no problem with the research policy or method. However, I would like you to consider the following points.

1. What is the significance of AQP4 being present in microgia in the early stages of development, and what does it mean that AQP4 is no longer positive at P20, when the brain is still maturing?

2. I understand that at 3 and 7 days after birth, the BBB is already forming to some extent. I understand that in CC in particular, the inflow of cells from blood vessels to the brain is somewhat restricted. In such a situation, what is the importance of AQP4 being transiently expressed in microgia?

3. When AQP4 is knocked out, the presence of CD11c-positive cells increases, but they still disappear at P20. In such a brain, for example, what kind of influence does it have on microglia after they become adults, and what kind of influence does it have on the whole brain or just the CC, or on that part of the brain?

4. The mechanism by which AQP4 affects CD11c-positive cells

Author Response

Reviewer 2.

We wish to thank this reviewer for the time she/he spent to evaluate our paper and for the nice words that were said about our work. The four points/questions raised by this reviewer are interesting reflexions that we will comment on below. These, rather than letting us introduce changes in the manuscript, make us think about future experiments and new lines of investigation. Only a very short sentence regarding the last question rise was introduced at the end of the discussion.

Minor points

1.- What is the significance of AQP4 being present in microgia in the early stages of development, and what does it mean that AQP4 is no longer positive at P20, when the brain is still maturing?

Answer: We need to start saying that microglia does not express AQP4. In the brain of neonate’s animals, we previously described (Mayo et al., IJMS-2023) that AQP4 abundantly express in astrocytes of corpus callosum (CC) by P7-P11, and later on by P20 the higher levels of AQP4 expression are over the end feet membrane of astrocytes surrounding the blood vessels in the brain cortex. What we show here in this work is that the specific neonate microglia CD11c+, appear in CC of WT and AQP4-KO animals, for a short window of time, and disappear later. These cells remain in the CC longer when the animal is nock out for AQP4, but we have never seen expression of AQP4 over microglia cells. The presence of this CD11c+ microglia have been associated with myelination, a process very active in CC during the first 10 days after birth, and after that this microglia subtype will go away from this tissue.

2.- . I understand that at 3 and 7 days after birth, the BBB is already forming to some extent. I understand that in CC in particular, the inflow of cells from blood vessels to the brain is somewhat restricted. In such a situation, what is the importance of AQP4 being transiently expressed in microglia?
Answer: Again, we must indicate to the Reviewer that we did not show, neither we have found, expression of AQP4 in microglia cells. What we reported here is a different time course of presence/expression of microglia CD11c+ cells in the corpus callosum of animals WT compare with animals that lack congenital expression of AQP4 in all tissues. We have seen transient presence/expression of microglia CD11c+ cells, but not of AQP4.

3.-. When AQP4 is knocked out, the presence of CD11c-positive cells increases, but they still disappear at P20. In such a brain, for example, what kind of influence does it have on microglia after they become adults, and what kind of influence does it have on the whole brain or just the CC, or on that part of the brain?

Answer: We believe or hypothesized that CD11c+ cells appear to be necessary for normal corpus callosum development during postnatal life. Moreover, the longer presence/expression of CD11c+ cells in CC of animals with AQP4 knocked out will help in this animal to obtain a normal development of the CC zone, that on the contrary situation, if these cells did not arrive and prolonged its expression in this tissue, an aberrant development of it will occur, carrying neural malformations in the animal.

  1. The mechanism by which AQP4 affects CD11c-positive cells

Answer: This is certainly a very important question that we are trying to address now in the laboratory, but we do not have yet a clear answer for it. We´ll keep working on it.

Round 2

Reviewer 1 Report

Comments and Suggestions for Authors

The authors improved the quality of the manuscript 

Reviewer 2 Report

Comments and Suggestions for Authors

The authors have made appropriate revisions to the previous comments. I feel that the quality of the paper has improved. I have no further comments to make.